# Statistically Valid Post-Deployment Monitoring Should Be Standard for AI-Based Digital Health

**Pavel Dolin, Weizhi Li, Gautam Dasarathy, Visar Berisha**
Arizona State University
Tempe, AZ
{pdolin, weizhili, gautamd, visar}@asu.edu

## Abstract

This position paper argues that post-deployment monitoring in clinical AI is under-developed and proposes statistically valid and label-efficient testing frameworks as a principled foundation for ensuring reliability and safety in real-world deployment. A recent review found that only 9% of FDA-registered AI-based healthcare tools include a post-deployment surveillance plan [1]. Existing monitoring approaches are often manual, sporadic, and reactive, making them ill-suited for the dynamic environments in which clinical models operate. We contend that post-deployment monitoring should be grounded in label-efficient and statistically valid testing frameworks, offering a principled alternative to current practices. We use the term "statistically valid" to refer to methods that provide explicit guarantees on error rates (e.g., Type I/II error), enable formal inference under pre-defined assumptions, and support reproducibility—features that align with regulatory requirements. Specifically, we propose that the detection of changes in the data and model performance degradation should be framed as distinct statistical hypothesis testing problems. Grounding monitoring in statistical rigor ensures a reproducible and scientifically sound basis for maintaining the reliability of clinical AI systems. Importantly, it also opens new research directions for the technical community—spanning theory, methods, and tools for statistically principled detection, attribution, and mitigation of post-deployment model failures in real-world settings.

## 1 Introduction

AI models play a growing role in healthcare by providing advanced tools for disease diagnosis, medical imaging analysis, treatment planning, and patient monitoring [2, 3, 4, 5, 6]. However, their promise is contingent on maintaining reliability and accuracy post-deployment—an area where significant challenges remain. AI-based digital health tools are known to experience performance degradation over time, which can have profound clinical implications, from missed diagnoses in radiology [3] to delayed interventions in critical care [5]. These declines are particularly pronounced in diverse healthcare settings, where variations in demographics, deployment sites, and equipment can exacerbate diagnostic disparities [7, 8, 9, 10], posing risks to patient safety.

Despite these risks, a recent study revealed that only 9% of FDA-registered AI-based healthcare tools include a post-deployment surveillance plan [1]. Yet, the FDA's guidelines for Software as a Medical Device (SaMD) [11] emphasize the importance of ongoing model evaluation, including the use of prospective, statistically valid real-world performance monitoring to ensure continued safety, effectiveness, and performance. Similarly, the National Institute of Standards and Technology (NIST) emphasizes post-deployment monitoring as a cornerstone of its AI Risk Management Framework, essential for managing risk and maintaining trust throughout the AI lifecycle [12]. In line with these

39th Conference on Neural Information Processing Systems (NeurIPS 2025) Position Paper Track.

expectations, it is critical to understand the mechanisms by which model performance can deteriorate after deployment in order to address the issue in a systematic and effective way.

Performance degradation in deployed AI models can arise from various sources: shifts in patient demographics, evolution of clinical practices, changes in medical equipment or protocols, emergence of new disease patterns, and variations in data acquisition procedures [8, 13, 9, 14, 15, 10]. At a high level, these data-related changes can be grouped into three categories: covariate shift [16, 17], label shift [18], and concept drift [19, 20, 21]. Covariate shift refers to changes in the input features while the relationship between the input features and the labels remains unchanged. For example, shifts in patient demographics and alterations in data collection methods. Label shift occurs due to the changes in the output distribution, while output-conditional feature distributions remain the same. For example, seasonal fluctuations in flu prevalence. On the other hand, concept drift refers to changes in the relationship between input features and labels, while the distribution of input features remains unchanged. This can occur due to shifts in clinical practice, new medical guidelines, changes in outcome prevalence, or the emergence of new confounding variables. Figure 1 (a), (b), and (c) depict simple examples of covariate shift, label shift, and concept drift, respectively, while Table 2 in the Appendix A summarizes key causes and examples in the clinical AI.

The described shifts in the data can make the model's previously learned associations less accurate or outright invalid. Although model performance can be evaluated to determine whether these shifts change performance meaningfully, common evaluation methods after deployment are often based on average performance metrics performed manually and sporadically by clinicians [13, 9]. By the time clinicians detect a decline in model performance, significant harm may have already occurred, and trust in the model may have been lost. Moreover, average performance metrics can mask degradation in specific patient subgroups [22]. Identifying and monitoring these subgroup-specific performance changes is crucial for ensuring effective care for all patients. However, effective and persistent post-deployment monitoring of this form is challenging due to the scarcity of ground truth labels [23, 24, 25].

While AI in healthcare includes both predictive and generative applications, this paper focuses exclusively on predictive models—such as those used for diagnosis, prognosis, and clinical risk scoring. We do not address generative models like large language models (LLMs). This focus is deliberate: even for predictive systems, post-deployment monitoring remains an unsolved challenge. Establishing rigorous methods in this domain is a necessary first step. **This paper argues that post-deployment monitoring remains underemphasized in the machine learning community—particularly in high-stakes applications like clinical AI, where errors can have severe consequences. Current practices are ad hoc, sporadic, and reactive, lacking the systematic rigor needed to ensure safety and reliability. We contend that integrating statistically valid testing frameworks into post-deployment workflows offers a principled and label-efficient foundation and should become a core component of the machine learning lifecycle for clinical applications.**

## 2 Related Work

**Data Shift Detection**    Covariate shift has been extensively studied in the machine learning community. Early theoretical work defined covariate shift and developed importance weighting techniques for adaptation [16, 17]. Subsequent research provided unified taxonomies of dataset shift types [26] and empirical studies evaluating drift detection methods in high-dimensional settings [27]. With respect to concept drift, [21] offers a comprehensive taxonomy of drift types and adaptation strategies in data-stream learning, focusing primarily on supervised and semi-supervised settings. [28] focuses on unsupervised scenarios where labels are scarce and categorizes detectors based on statistical, clustering, and reconstruction principles. In the context of our work, [29] introduces a classifier-independent drift detector based on hierarchical hypothesis testing, one of the few existing approaches that aligns with the statistically principled framework we advocate. [30] highlights the risks of dataset shift in deployed clinical ML systems and advocates for clinician-in-the-loop monitoring to detect and respond to data shifts, emphasizing governance and oversight.

A range of statistical tests can be applied to compare pre- and post-deployment distributions, enabling formal detection of a shift through two-sample hypothesis testing. Table 1 summarizes common parametric and nonparametric tests, which can be applied to both data-shift detection [27, 31] and model-performance monitoring [32].

**Model Performance Monitoring** At deployment scale, many concurrent monitors (features, subgroups, metrics) benefit from online multiple-testing with anytime-valid FDR control (SAFFRON) and e-value procedures (e-BH) for adaptive false-alarm control [33, 34]. In healthcare applications, [35] demonstrated how shifts in patient demographics and clinical workflows can degrade predictive-model performance, while [36] proposed practical evaluation strategies under distributional shift, highlighting challenges particularly relevant to clinical deployments. [37] used CUSUM control charts to track input drift in medical AI, and [38] applied statistical process control methods for radiological data monitoring. [39] focused on detecting calibration drift in predictive models, whereas [40] introduced adaptive windowing for real-time multimodal performance monitoring.

**MLOps** Finally, the operationalization of AI model monitoring has been advanced by the field of *MLOps*. While our focus is on statistically valid monitoring, there has been considerable development in system infrastructure. The work in [41] proposed a comprehensive MLHOps framework, detailing deployment pipelines and monitoring components tailored for healthcare AI. Similarly, [42] identified architectural considerations and practical challenges for real-world post-deployment monitoring. This position paper complements these system-level frameworks by arguing for a statistically principled foundation for model monitoring, one that is label-efficient, interpretable, and aligned with regulatory expectations. We highlight how framing monitoring tasks as hypothesis testing problems enables systematic and actionable approaches that can be integrated into existing MLOps pipelines to enable safe deployment.

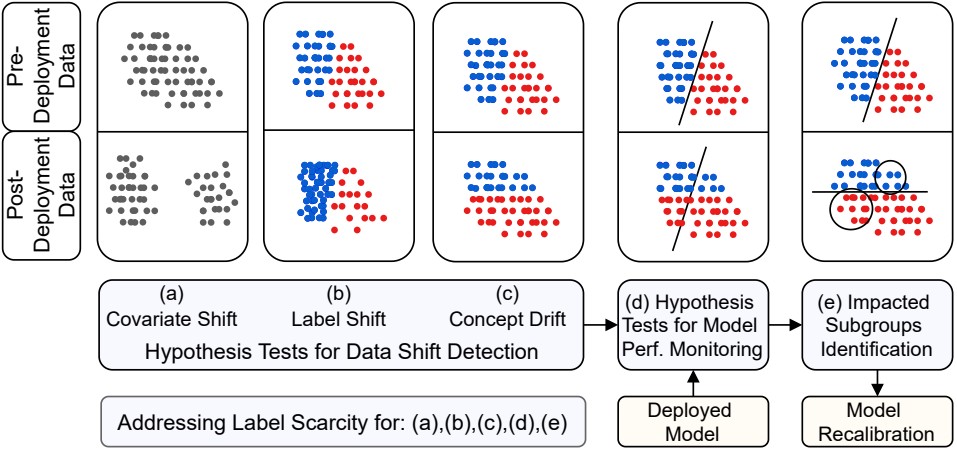

Figure 1: Framing Post-Deployment Monitoring as Hypothesis Testing. Binary classification example. A hypothesis test for (a) covariate shift—input features distribution changes, but the relationship between labels and input features remains the same, (b) label shift—output distribution changes, while output-conditional feature distributions remain the same, (c) concept drift—relationship between labels and input features changes, but the distribution of the input features remains the same. If a statistically significant change is observed, a hypothesis test for (d) model performance degradation is performed. If a model is affected by the change, (e) impacted subgroup identification is performed and used for target label collection and model recalibration. One of the open problems is addressing label scarcity for each of the described stages.

## 3 Framing Post-Deployment Monitoring as Hypothesis Testing

Given the challenges outlined in the introduction—manual monitoring, reliance on coarse average metrics, label scarcity, and FDA expectations for statistical validity—there is a need for rigorous and scalable monitoring tools in clinical AI. We propose reframing post-deployment monitoring as a collection of statistically grounded two-sample hypothesis tests. This framing enables formal decision-making with controlled error rates, moving beyond heuristic-based methods that lack statistical guarantees. It also provides a principled foundation for aligning model monitoring with regulatory standards and clinical safety needs.

To this end, we organize post-deployment monitoring into two core stages: (I) *Data Shift Detection* and (II) *Model Performance Monitoring*. Within the data shift stage, we introduce hypothesis tests for covariate shift (Section 4.1) and concept drift (Section 4.2). In the model performance stage, we develop tests for detecting degradation in overall model accuracy (Section 5.1) as well as distributional shifts in prediction correctness - e.g. do errors concentrate in specific subgroups? (Section 5.2). Each monitoring task is cast as a two-sample hypothesis test comparing pre- and post-deployment distributions, providing statistically grounded monitoring, as depicted in Figure 1. The post-deployment monitoring is modality-agnostic, as long as the data can be represented as input features and clinical/demographic variables.

While conceptually straightforward, this formulation exposes several unresolved challenges. Most notably, evaluating performance degradation requires access to post-deployment ground truth labels, which are often delayed, costly, or entirely unavailable. In high-dimensional settings, required sample sizes rise and test power can drop; thus, label-efficient tactics (e.g., sequential looks and active labeling) and explicit reporting of decision margins/thresholds are essential. This motivates research into label-efficient monitoring strategies, including active learning, surrogate labeling, weak supervision, and model-based uncertainty estimation. Moreover, once degradation is detected, identifying the most affected subgroups remains an open problem critical to ensuring fairness and guiding retraining. Throughout the manuscript, we provide formal problem statements and examples of existing approaches and present these challenges as open problems (OP) for the community.

Let $\mathcal{D}_{t_0}$ and $\mathcal{D}_{t_1}$ denote i.i.d. samples [1] collected at pre-deployment and post-deployment time points $t_0$ and $t_1$, drawn from distributions $p_{t_0}$ and $p_{t_1}$, respectively. Let $f : \mathbb{R}^d \to \mathbb{R}$ be a model mapping clinical inputs to predictions, and let $M_t = g(f, p_t)$ denote its performance under distribution $p_t$. We frame monitoring tasks as two classes of two-sample tests:

**(I) Data Shift Detection** A key objective in post-deployment monitoring is to determine whether the distribution of patient characteristics has changed significantly between $t_0$ and $t_1$. We use this test type to detect distributional shifts, as detailed in Sections 4.1 and 4.2. To this end, we define the null and alternative hypotheses, $H_0$ and $H_1$, as follows:

$$\begin{aligned} H_0 &: p_{t_0} = p_{t_1} \\ H_1 &: p_{t_0} \neq p_{t_1} \end{aligned} \tag{1}$$

**(II) Model Performance Monitoring** Another important objective is to detect the performance degradation of the AI model after deployment. Given a user-specified performance evaluation function $g$ (e.g., classification accuracy), we compare the model's performance over the data distributions at $t_0$ and $t_1$. We use this test to detect performance degradation as detailed in Section 5. Given $\tau > 0$, a user-defined threshold for acceptable performance degradation [2], the corresponding hypotheses are:

$$\begin{aligned} H_0 &: M_{t_0} - M_{t_1} \leq \tau \\ H_1 &: M_{t_0} - M_{t_1} > \tau \end{aligned} \tag{2}$$

In both cases, we compute a test statistic $T(\mathcal{D}_{t_0}, \mathcal{D}_{t_1})$ and compare it to a critical value $c$ to determine whether the observed difference is statistically significant:

$$\begin{aligned} \text{Reject } H_0 : \quad &\text{if} \quad T(\mathcal{D}_{t_0}, \mathcal{D}_{t_1}) > c \\ \text{Accept } H_0 : \quad &\text{otherwise.} \end{aligned} \tag{3}$$

Table 1 summarizes candidate two-sample test statistics—parametric and non-parametric—and the associated assumptions and power trade-offs. Appendix D outlines our test-selection rationale, and Appendices F and G describe the parametric and non-parametric tests, respectively.

## 4 Data Shift Detection

One of the initial challenges in post-deployment monitoring is to detect data-only distributional changes between the pre- and post-deployment time points, denoted $t_0$ and $t_1$. We distinguish

---

[1] i.i.d. assumption is local within each period (samples at $t_0$ or $t_1$) but not across time.

[2] The threshold is set based on application-specific considerations that reflect what constitutes a clinically-meaningful drop in performance.

| | | Tests | When to Use / Notes | Data Distribution Assumptions |
|---|---|---|---|---|
| **Parametric** | **M** | z test | When population standard deviation is known | normality, known variance |
| | | Two-Sample t-Test | When variances are unknown but assumed equal | normality, equal variances |
| | | Welch's t-Test | When variances are unknown and potentially unequal | normality, unequal variances |
| | **V** | F-Test | Compare two variances; affected by skewness | normality |
| | | Bartlett's Test | Extends F-Test to multiple groups; more stable than F-test | normality |
| **Non-Parametric** | **M** | Mann-Whitney U Test | When distribution shape is unknown or non-normal | |
| | **V** | Levene's Test | When normality is uncertain | |
| | **Distr. Shifts** | Kolmogorov-Smirnov (KS) Test | General-purpose test; best for moderate sample sizes | |
| | | Anderson-Darling Test | When identifying shifts in rare events is critical; better than KS test for tail differences | |
| | | Friedman-Rafsky Test | Uses graph-based approach using minimum spanning tree | |
| | | Maximum Mean Discrepancy (MMD) | Effective for detecting shifts in high-dimensional data, needs an appropriate kernel choice | |

Table 1: Summary of two sample test statistics for detecting differences between $p_{t_0}$ and $p_{t_1}$, including assumptions and use cases. All methods assume *i.i.d.* data. Note: "M" denotes mean, "V" denotes variance

between three types of data shift: *covariate shift*, where the distribution of inputs changes while the input-output relationship remains fixed, *label shift*, where the marginal distribution of outputs changes while the distribution of inputs given outputs remains fixed; and *concept drift*, where the conditional relationship between inputs and outputs changes, while the input distribution remains fixed [26]. Figure 1 (a), (b), and (c) illustrate a simple example of covariate shift, label shift, and concept drift. In the following subsections, we define covariate shift and concept drift and develop hypothesis tests to detect them. We defer label shift to Appendix B, since its testing procedure is analogous to the covariate shift test.

Let $(\mathbf{S}, \mathbf{C})$ denote a pair of random variables in the sample space $\mathcal{S} \times \mathcal{C}$, representing input and clinical feature variables, respectively. We separate $\mathbf{S}$ from $\mathbf{C}$ to reflect their distinct roles in the monitoring pipeline. The model operates solely on $\mathbf{S}$, which we assume encodes all relevant information for prediction, while $\mathbf{C}$ is treated as a latent variable that can be used in the identification of impacted subgroups. We use $Y \in \{0, 1\}$ to denote the corresponding label random variable for $(\mathbf{S}, \mathbf{C})$, resulting in the tuple $(\mathbf{S}, \mathbf{C}, Y)$, which includes both feature and label variables. Assuming that $\{(\mathbf{S}, \mathbf{C}, Y)_i\}_{i=1}^n$ are *i.i.d.*, we express the marginal distribution of $(\mathbf{S}, \mathbf{C}, Y)$ as $p(\mathbf{s}, \mathbf{c}, y) = p(y \mid \mathbf{s}, \mathbf{c}) p(\mathbf{s}, \mathbf{c})$, where $\mathbf{s}$, $\mathbf{c}$ and $y$ denote the realizations of the random variables.

## 4.1 Covariate Shift

Following the definition in [26], covariate shift corresponds to the case where $p_{t_1}(\mathbf{s}, \mathbf{c}) \neq p_{t_0}(\mathbf{s}, \mathbf{c})$, while $\forall (\mathbf{s}, \mathbf{c}) \in \mathcal{S} \times \mathcal{C}$, $p_{t_1}(y \mid \mathbf{s}, \mathbf{c}) = p_{t_0}(y \mid \mathbf{s}, \mathbf{c})$. To formalize covariate shift detection as a statistical hypothesis test, let $\mathcal{D}_{t_0}^{sc} = \{(\mathbf{S}, \mathbf{C})_i\}_{i=1}^{n_{t_0}}$ and $\mathcal{D}_{t_1}^{sc} = \{(\mathbf{S}, \mathbf{C})_i\}_{i=1}^{n_{t_1}}$ denote two sets of *i.i.d.* data collected at $t_0$ and $t_1$. We denote $p_{t_0}(\mathbf{s}, \mathbf{c})$ and $p_{t_1}(\mathbf{s}, \mathbf{c})$ as the joint distributions of the input and clinical features at the pre- and post-deployment stages. The goal is to monitor whether a statistically significant change has occurred in this joint distribution of covariates $(\mathbf{S}, \mathbf{C})$, which can be posed as the following two-sample hypothesis testing problem:

$$
\begin{aligned}
H_0 : \quad & p_{t_0}(\mathbf{s}, \mathbf{c}) = p_{t_1}(\mathbf{s}, \mathbf{c}), \\
H_1 : \quad & p_{t_0}(\mathbf{s}, \mathbf{c}) \neq p_{t_1}(\mathbf{s}, \mathbf{c}).
\end{aligned}
\tag{4}
$$

We note that, for $H_0$ and $H_1$, and output label $y$, this test assumes that the conditional distribution of $y$ remains unchanged—that is, $p_{t_0}(y|\mathbf{s}, \mathbf{c}) = p_{t_1}(y|\mathbf{s}, \mathbf{c})$ holds $\forall (\mathbf{s}, \mathbf{c}) \in \mathcal{S} \times \mathcal{C}$. Under the null hypothesis $H_0$, the joint distribution of features and clinical variables remains unchanged between $t_0$ and $t_1$; that is, $p_{t_0}(\mathbf{s}, \mathbf{c}) = p_{t_1}(\mathbf{s}, \mathbf{c}), \forall (\mathbf{s}, \mathbf{c}) \in \mathcal{S} \times \mathcal{C}$. The alternative hypothesis $H_1$ posits that $p_{t_0}(\mathbf{s}, \mathbf{c}) \neq p_{t_1}(\mathbf{s}, \mathbf{c}), \exists (\mathbf{s}, \mathbf{c}) \in \mathcal{S} \times \mathcal{C}$, indicating a potential shift in the underlying distribution of patient covariates.

The samples $\mathcal{D}_{t_0}^{sc}$ and $\mathcal{D}_{t_1}^{sc}$ are compared using two-sample hypothesis tests to decide between $H_0$ and $H_1$. The choice of test depends on the monitoring objectives and assumptions about the data. A variety of parametric and non-parametric methods can be applied, which are reviewed in Appendix D.

**OP: Choosing an Appropriate Test for High-Dimensional Data**    Several challenges remain in the practical application of covariate shift testing. First, evaluating and selecting an appropriate test for a given problem is nontrivial, particularly in high-dimensional settings where testing power can be low. The choice of a test can drastically impact sensitivity and interoperability. Second, covariate shift tests rely on the assumption that the conditional distribution $p(y \mid \mathbf{s}, \mathbf{c})$ remains unchanged. This assumption is typically unverifiable in practice without explicit testing and may be violated, undermining the validity of the test. Robust approaches that can either test for this invariance or remain effective under its relaxation are an important direction for future work.

## 4.2   Concept Drift

Following the definition in [26], concept drift corresponds to: $p_{t_1}(y \mid \mathbf{s}, \mathbf{c}) \neq p_{t_0}(y \mid \mathbf{s}, \mathbf{c}), \exists (\mathbf{s}, \mathbf{c}) \in \mathcal{S} \times \mathcal{C}$ while $p_{t_1}(\mathbf{s}, \mathbf{c}) = p_{t_0}(\mathbf{s}, \mathbf{c}), \forall (\mathbf{s}, \mathbf{c}) \in \mathcal{S} \times \mathcal{C}$. We define $\mathcal{D}_{t_0}^{scy} = \{(\mathbf{S}, \mathbf{C}, Y)_i\}_{i=1}^{n_0}$ and $\mathcal{D}_{t_1}^{scy} = \{(\mathbf{S}, \mathbf{C}, Y)_i\}_{i=1}^{n_1}$ to represent the collection of covariates and true labels at $t_0$ and $t_1$. Furthermore, let $p_{t_0}(\mathbf{s}, \mathbf{c}, y)$ and $p_{t_1}(\mathbf{s}, \mathbf{c}, y)$ denote the joint distributions of $(\mathbf{S}, \mathbf{C}, Y)$ at $t_0$ and $t_1$, for which the pre- and post-deployment datasets $\mathcal{D}_{t_0}^{scy}$ and $\mathcal{D}_{t_1}^{scy}$ are sampled. We formulate the two-sample hypothesis testing problem to detect the concept drift as:

$$
\begin{aligned}
H_0: & \quad p_{t_0}(\mathbf{s}, \mathbf{c}, y) = p_{t_1}(\mathbf{s}, \mathbf{c}, y) \\
H_1: & \quad p_{t_0}(\mathbf{s}, \mathbf{c}, y) \neq p_{t_1}(\mathbf{s}, \mathbf{c}, y).
\end{aligned}
\tag{5}
$$

We note that this test assumes that the joint distributions of the features and clinical/demographic variables at the pre- and post-deployment stages remain unchanged, that is $p_{t_0}(\mathbf{s}, \mathbf{c}) = p_{t_1}(\mathbf{s}, \mathbf{c})$ holds $\forall (\mathbf{s}, \mathbf{c}) \in \mathcal{S} \times \mathcal{C}$. Under $H_0$, the joint distribution of features, clinical/demographic, and label variables remains consistent over time, indicating that the model's performance has not degraded. Under $H_1$, the joint distribution of inputs and labels has changed, which may affect the model's performance if it is sensitive to such distribution shifts. Similar to the covariate shift discussed in Section 4.1, the task of monitoring concept drift is framed as detecting distributional shifts. This involves comparing the datasets $\mathcal{D}_{t_0}^{scy}$ and $\mathcal{D}_{t_1}^{scy}$ using a two-sample test to determine whether to reject $H_0$. We refer readers to the two-sample tests described in Appendix D for further details.

**OP: Drift Mechanism Attribution**    While concept drift is formally defined as changes in $p(y \mid \mathbf{s}, \mathbf{c})$, standard two-sample tests cannot directly test conditional distributions. We employ sequential testing [43]: first test for label shift (Appendix B); if absent, test the joint distribution $p(\mathbf{s}, \mathbf{c}, y)$ for concept drift. When both shifts co-occur, distinguishing the mechanism requires additional methods. Class-conditional testing, separately test whether $p(\mathbf{s}, \mathbf{c} \mid y)$ has changed for each outcome class, as label shift preserves these distributions while concept drift does not, though this requires sufficient labeled samples per class. Black Box Shift Estimation (BBSE) [43] can estimate the magnitude of label shift from unlabeled data, providing qualitative insight into the relative contribution of each mechanism. Alternative approaches like stratified binning or kernel methods [44] are possible but remain impractical due to the curse of dimensionality.

**OP: Addressing Label Scarcity**    Evaluating the hypothesis test requires post-deployment ground truth labels; however, these are costly and time-consuming to obtain. To mitigate this, we need new approaches for surrogate models to approximate the true label $y$ in both pre- and post-deployment settings, building on the foundation of surrogate endpoints established in clinical trials [45]. In healthcare machine learning, surrogate labels are often derived from data correlated with clinical outcomes, such as billing codes [46], lab results [47], or earlier outcomes like 30-day readmission used in place of 90-day outcomes [48]. Alternatively, surrogate labels can be generated from combinations of weak sources—heuristics, knowledge bases, or auxiliary models [49, 50]. While practical in label-scarce settings, these proxies may introduce noise or degrade over time, limiting their reliability [51, 52, 53]. This can be formulated as a regression problem $y = J(\mathbf{s}, \mathbf{c}, \hat{y}) + \epsilon_i$, where $\hat{y}$ is model's prediction, $y$ is a label, $\epsilon_i$ is the residual noise, and $J$ represents the function approximated by the surrogate model. $J$ can be estimated by the Prediction Aided by Surrogate Training (PAST) algorithm [54]. Beyond surrogate modeling, this setting motivates new directions in label-efficient hypothesis testing—an area that remains underexplored. For example, Li et al. [55, 23, 56] propose a query strategies for active labeling of samples in two-sample tests. They show this preserves validity and enhances test power under label constraints.

# 5 Model Performance Monitoring

In this section, we present two complementary approaches to monitoring model performance, each framed as a two-sample hypothesis test. Subsection 5.1 introduces a test for monitoring changes in the model's performance score, while Subsection 5.2 describes a test for monitoring shifts in the distribution of prediction correctness.

## 5.1 Monitoring Performance Score

Direct assessment of the performance for an AI model $\hat{Y} = f(\mathbf{S})$, $(\mathbf{S}, \mathbf{C}) \sim p(\mathbf{s}, \mathbf{c})$, is essential for detecting degradation. To do that, one needs to collect the true label $Y$ at the pre- and post-deployment stages, $t_0$ and $t_1$, and evaluate the performance score using them along with the prediction variable $\hat{Y}$. Specifically, we write $M = g(f, p(\mathbf{s}, \mathbf{c}, y))$ to denote an evaluation function that outputs the performance score, or metric $M$, with respect to the model $f$ and the data distribution $p(\mathbf{s}, \mathbf{c}, y)$.

There are many choices for the evaluation function $g$. Appendix C describes performance metrics that are commonly used. For instance, let $M_{t_1}$ (or $M_{t_0}$) denote the performance score of a model at $t_1$ (or $t_0$). If we select classification accuracy for $M_{t_1} = g(f, p_{t_1})$, we then have $M_{t_1} = \int \int \int \mathbb{1}_{f(\mathbf{s})=y} p_{t_1}(\mathbf{s}, \mathbf{c}, y) \, d\mathbf{s} d\mathbf{c} dy$. Typically, one does not have access to $p_{t_0}$ or $p_{t_1}$ to evaluate $M_{t_0}$ or $M_{t_1}$; instead, one resorts to computing the empirical metric, e.g., $\hat{M}_{t_1} = \frac{1}{|\mathcal{D}_{t_1}^{scy}|} \sum_{(\mathbf{s},\mathbf{c},y) \in \mathcal{D}_{t_1}^{scy}} \mathbb{1}_{f(\mathbf{s})=y}$, where $\mathcal{D}_{t_1}^{scy} = \{(\mathbf{s}, \mathbf{c}, y)_i\}_{i=1}^{n_1}$.

To this end, we establish two complementary two-sample tests for monitoring the performance score: (1) performance degradation relative to pre-deployment performance, and (2) specification threshold testing, as described below.

**Performance Deviation Testing**   Implemented through one-sided tests (OST), performance deviation testing assesses whether model performance has remained stable within an acceptable margin relative to its pre-deployment baseline. This approach is particularly useful for demonstrating sustained performance rather than merely detecting degradation. To this end, we formalize the detection of performance deviation as a one-sided two-sample testing problem:

$$
\begin{aligned}
H_0 : \quad & M_{t_0} - M_{t_1} \leq \tau_{\text{deg}}, \\
H_1 : \quad & M_{t_0} - M_{t_1} > \tau_{\text{deg}}
\end{aligned}
\tag{6}
$$

where $\tau_{\text{deg}} > 0$ denotes a predefined threshold representing the maximum tolerable performance degradation. The decision between $H_0$ and $H_1$ is made by comparing $\hat{M}_{t_0}$ and $\hat{M}_{t_1}$, computed from the pre- and post-deployment datasets $\mathcal{D}_{t_0}^{scy}$ and $\mathcal{D}_{t_1}^{scy}$, respectively.

**Specification Threshold Testing**   directly evaluates whether the current model performance meets predetermined minimum requirements, which is important for regulatory compliance and clinical safety standards. In contrast to performance deviation testing, specification threshold testing assesses only whether the post-deployment performance score $M_{t_1}$ falls below a predefined threshold $\tau_{\text{spec}} > 0$. The goal is to verify compliance with the specified performance standards. Formally, we have

$$
\begin{aligned}
H_0 : \quad & M_{t_1} \geq \tau_{\text{spec}}, \\
H_1 : \quad & M_{t_1} < \tau_{\text{spec}}.
\end{aligned}
\tag{7}
$$

The decision between $H_0$ and $H_1$ is based on evaluating the empirical score $\hat{M}_{t_1}$ using the dataset $\mathcal{D}_{t_1}^{scy}$.

**OP: Impacted Subgroups Identification**   The performance metrics and hypothesis tests presented above capture average performance over the entire distribution $p(\mathbf{s}, \mathbf{c}, y)$, overlooking significant performance variations across different subgroups. Inspired by Cohort Enrichment [3] [57] and Exceptional Model Mining (EMM) [4] [58] strategies, one can systematically identify subgroups

---

[3]Cohort Enrichment refers to identifying subsets of the data where a particular phenomenon (e.g., degradation) is amplified relative to the general population

[4]EMM is a generalization of subgroup discovery aimed at finding subpopulations where model behavior deviates significantly from the global norm—whether in performance, fairness, or other metrics

experiencing meaningful performance decline. We define the subgroup-specific performance as $M_t^{\mathcal{G}} = g\left(f, p_t\left(\mathbf{s}, \mathbf{c}, y \mid \mathcal{G}\right)\right)$ for a candidate subgroup $\mathcal{G} \subseteq \mathcal{S} \times \mathcal{C}$. The task of identifying subgroups with the most greatest performance degradation between $t_0$ and $t_1$ is then formalized as the following optimization problem: $\max_{\mathcal{G} \subseteq \mathcal{S} \times \mathcal{C}} M_{t_0}^{\mathcal{G}} - M_{t_1}^{\mathcal{G}}$, s.t. $|\mathcal{G}| \geq r$, where $r$ is a predefined minimum group size. Solving it offers valuable insights for: (1) identifying features where the model's discriminative power has shifted, (2) detecting subgroups that experience disproportionate performance degradation, and (3) uncovering complex interaction patterns that may signal vulnerable populations.

**OP: Addressing Label Scarcity**  A separate potential direction on addressing label scarcity is rooted in active learning [59, 60, 61, 62], which aims to develop classification models under limited label availability. In the context of our work, active learning offers a principled approach to selecting which covariate instances $(\mathbf{s}, \mathbf{c})$ should be labeled, thereby improving the performance of the model $f$ efficiently. Typically, this involves constructing an acquisition function $q(\mathbf{s}, \mathbf{c})$ that quantifies the informativeness of instances across the covariate space $\mathcal{S} \times \mathcal{C}$. The instance with the highest acquisition score, $\arg\max_{(\mathbf{s}, \mathbf{c}) \in \mathcal{S} \times \mathcal{C}} q(\mathbf{s}, \mathbf{c})$, is selected for labeling and used to update the model $f$. Representative acquisition functions include ensemble-based uncertainty estimation techniques such as Query-by-Committee (QBC)[63] and deep ensembles[64]. Intuitively, these functions measure the uncertainty of model predictions over the covariate space. Regions with the highest predictive uncertainty often correspond to areas where the model underperforms. Consequently, prioritizing label queries in these regions facilitates more effective detection of model degradation.

## 5.2 Monitoring Prediction Correctness

This section introduces the concept of detecting changes in the joint distribution of model features and prediction correctness. The performance score monitoring, presented in Section 5.1, evaluates whether there is overall performance degradation across the entire population. In contrast, the method described below is designed to detect performance changes even within a local subpopulation during post-deployment monitoring. This is achieved by framing post-deployment monitoring as a two-sample testing problem for identifying distributional shifts, rather than differences in average performance. Before formalizing this approach, we introduce the correctness indicator $Z \in \{0, 1\}$ as follows:

$$Z = \begin{cases} 1, & \text{if } \hat{y} = y \quad \text{(correct prediction)}, \\ 0, & \text{if } \hat{y} \neq y \quad \text{(incorrect prediction)} \end{cases} \tag{8}$$

where $\hat{y} = f(\mathbf{s})$ denotes the model's prediction. Herein, we also reuse the notations of pre- and post-deployment data, and define $\mathcal{D}_{t_0}^{scz} = \{(\mathbf{S}, \mathbf{C}, Z)_i\}_{i=1}^{n_0}$ and $\mathcal{D}_{t_1}^{scz} = \{(\mathbf{S}, \mathbf{C}, Z)_i\}_{i=1}^{n_1}$ to represent the collection of covariates and model correctness indicators at $t_0$ and $t_1$. Furthermore, let $p_{t_0}(\mathbf{s}, \mathbf{c}, z)$ and $p_{t_1}(\mathbf{s}, \mathbf{c}, z)$ denote the joint distributions of $(\mathbf{S}, \mathbf{C}, Z)$ at $t_0$ and $t_1$, for which the pre- and post-deployment datasets $\mathcal{D}_{t_0}^{scz}$ and $\mathcal{D}_{t_1}^{scz}$ are sampled. We formulate the following hypothesis test to detect distribution shifts in the model's predictions:

$$\begin{aligned} H_0: & \quad p_{t_0}(\mathbf{s}, \mathbf{c}, z) = p_{t_1}(\mathbf{s}, \mathbf{c}, z) \\ H_1: & \quad p_{t_0}(\mathbf{s}, \mathbf{c}, z) \neq p_{t_1}(\mathbf{s}, \mathbf{c}, z). \end{aligned} \tag{9}$$

The nonparametric two-sample tests described in Appendix G for detecting distributional shifts can be applied to evaluate the hypothesis from $\mathcal{D}_{t_0}^{scz}$ and $\mathcal{D}_{t_1}^{scz}$.

This approach provides a distinct advantage over average performance score monitoring, as discussed in Section 5.1. For instance, it can alert users to significant shifts in model performance within specific regions of the covariate space, even when the model's overall performance remains stable.

**OP: Identifying Impacted Subgroups**  Similar to the open problems in the previous subsection, instead of focusing on the model's overall performance, we can also identify subgroups responsible for performance decline in the case of joint distribution. By selecting a discrepancy function $\Delta(p_{t_0}, p_{t_1})$—where $\Delta$ denotes a measure of discrepancy, such as an $f$-divergence [65]—to quantify the difference between distributions, we can formulate the task of identifying subgroups that exhibit distributional differences as the following optimization problem:$\max_{\mathcal{G} \subseteq \mathcal{S} \times \mathcal{C}} \Delta\left(p_{t_0}(\mathbf{s}, \mathbf{c}, z \mid \mathcal{G}), p_{t_1}(\mathbf{s}, \mathbf{c}, z \mid \mathcal{G})\right)$, s.t. $|\mathcal{G}| \geq r$, where $r$ denotes the minimum size of the subgroup, pre-specified based on clinical considerations. Solving this optimization

problem yields a subgroup $\mathcal{G}$ of size at least $r$ that maximizes the performance-related distributional discrepancy between $t_0$ and $t_1$.

**OP: Detecting Subtle Shifts**  Lastly, performance drift often unfolds gradually rather than through abrupt shifts. Given limited data, designing statistically rigorous and sensitive tests that can detect such gradual degradation—especially within specific patient subgroups—remains an important open challenge. Promising directions include the use of active two-sample testing strategies [23], which adaptively select informative samples to boost test power under label scarcity, as well as adaptive windowing and monitoring methods [39, 40] that track cumulative changes over time and are well-suited for detecting subtle shifts.

# 6   Alternative Views

Several well–established research threads could serve as alternatives to the post-deployment monitoring problem. We examine the pros and cons of the top three alternatives to our proposed approach: *continual learning*, *Bayesian change–point detection*, and *conformal-prediction–based monitoring*.

**Continual Learning**  or lifelong learning algorithms update model parameters online to accommodate non-stationary data distributions [66, 67, 68]. Advantages include low latency adaptation as well as theoretical guarantees. Models can react to drift on the very next batch, which is attractive when label feedback is cheap. Additionally, PAC-Bayesian [69] or regret bounds are available for certain online update rules. However, this comes with several limitations: label requirement, auditability and traceability and silent failure. State-of-the-art continual learners still rely on frequent ground-truth labels to avoid catastrophic forgetting [70]. In clinical applications, those labels are costly or delayed. Regulatory guidelines (e.g. FDA SaMD) require reproducible model versions. Online updates create a moving target that complicates version control, performance analysis, and root-cause analysis [11]. Finally, without an external test, online updates can chase noisy fluctuations and cause silent accuracy drops [71], which in turn may widen performance gaps for under-represented sub-groups in the data [72].

**Bayesian Change–Point Detection (BOCPD)**  offers a probabilistic approach by maintaining a posterior distribution over run lengths and updating this belief as new data stream in [73, 74]. This method provides coherent uncertainty quantification, allowing drift detection systems to trigger alarms based on posterior probabilities. It is also sequentially efficient: when using conjugate-exponential models, updates can be computed in constant amortized time per observation. However, BOCPD relies heavily on calibrated priors, which are rarely available or reliable in clinical contexts. Furthermore, applying BOCPD to high-dimensional data often requires approximate inference via particle filters or sequential Monte Carlo methods, which may be computationally infeasible for hospital-scale EHR feeds. Importantly, while BOCPD indicates *when* a change occurred, it does not identify *where* the change took place—limiting its usefulness in root-cause analysis and mitigation planning.

**Conformal Prediction and Exchangeability Martingales**  offers a distribution-free framework for post-deployment monitoring. These methods maintain finite-sample validity under the assumption of exchangeability and have been extended to detect drift by monitoring conformal $p$-values or exchangeability martingales [75, 76, 77]. They are attractive in that they do not require strong parametric assumptions, and variants such as Mondrian or conditional conformal predictors can operate effectively using weak labels. Still, this approach has notable weaknesses. The assumption of exchangeability is fragile in real-world settings like healthcare, where temporal, site-level, and treatment-based correlations are pervasive and violate i.i.d. conditions [78]. Furthermore, the signal provided by conformal methods is often blunt: they flag distributional shift only after prediction sets inflate, and do not distinguish between covariate and concept drift, nor do they identify the specific subgroups affected. As such, they tend to be reactive rather than proactive—triggering alarms after performance degrades, rather than before.

**Summary**  Two-sample hypothesis testing combines rigorous error control, label efficiency, and interpretability. It provides explicit $\alpha$-level guarantees on type I error and power-based control of type II error. Unlike conformal methods, label-free tests such as kernel MMD [79] compare input

distributions directly and highlight the feature regions driving the difference, providing regulator-friendly, interpretable evidence of shift. Detection is also modular: once drift is identified, retraining, recalibration, or continual learning can follow as appropriate. In contrast, continual learning, Bayesian monitoring, and conformal prediction each address part of the monitoring problem but fall short on one or more axes—supervision cost, statistical guarantees, or auditability. Hypothesis testing, by covering all three, is the most robust and regulator-ready foundation for monitoring AI systems in healthcare.

# 7    Limitations and Future Directions

This position paper focuses on statistically valid, label-efficient post-deployment monitoring and intentionally stops short of a fully causal treatment of failure attribution or label-semantics dynamics; we discuss monitoring of surrogate-label drift but leave causal analyses to future work. While we do not assume i.i.d. behavior across time (i.e., we assume that the data distribution has shifted relative to pre-deployment), we do assume that data are locally i.i.d. within pre- and post-deployment windows; extending the framework to handle explicit temporal dependence, time-series change detection, and broader nonstationarity is a natural next step.

While our formulation is modality-agnostic—operating on learned embeddings and applying the same drift, performance, and subgroup tests—we note a trade-off: higher dimensionality increases sample requirements and can reduce power. Consequently, label-efficient tactics (e.g., sequential looks and active labeling) together with clear reporting of margins and thresholds are key.

In the paper, we present statistical tests with binary outcomes. A natural extension is to treat validity as a continuum shaped by Pareto trade-offs among false-alarm control, detection delay, label budget, and subgroup granularity. Different operating points allocate limited risk and labeling resources differently across populations and time, tracing a frontier of feasible guarantees. This mirrors fairness–accuracy trade-offs in risk scoring: incompatible desiderata cannot be simultaneously optimized, so practitioners must select context-specific operating points with explicit, transparent priorities [80].

While our work focuses on data- and model-level monitoring, deployments should also track downstream clinical impact—e.g., shifts in treatment patterns, workflow latency, and patient outcomes—using pragmatic designs with governance-backed thresholds; this operational layer is complementary to our scope. Finally, while we present a traditional ML formulation, emerging generative tools (e.g., LLMs, diffusion models, agents, and synthetic-data pipelines) introduce additional monitoring challenges—including prompt and data-provenance drift, stochastic output variability and hallucinations, content and usage safety, and generator–downstream feedback loops—that our framework does not yet cover. Extending statistically valid monitoring to these generative settings is an important direction for future work.

# 8    Conclusion

In this paper, we have presented a statistical framework for monitoring the performance of AI-based digital health technologies post-deployment. By framing performance degradation detection as a series of hypothesis testing problems, we provide rigorous methods for identifying distributional shifts and model performance degradation, and we pose several open problems, notably in addressing label scarcity and impacted subgroup identification. Our approach enables statistically grounded, evidence-based criteria for detecting when intervention is needed, reduces reliance on subjective assessments, and facilitates targeted performance analysis across different patient populations, while also aligning with the FDA's approach to performance evaluation.

## Acknowledgments

This work was supported in part by the Office of Naval Research (ONR) under grant N00014-21-1-2615, the John and Tami Marick Foundation, and the National Science Foundation (NSF) under grant CCF-2048223.

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

## A  Reasons for the Data Shift

| | Reason | Example |
|---|---|---|
| **Covariate Shift** | Changes in Demographics | Changes in patient demographics alter input features like age or comorbidities [81] |
| | Data Quality Issues | New EHR system leads to missing test results or erroneous data entry [81] |
| | Changes in Data Collection Methods | Transitioning to a new system changes lab result units or reference ranges [82] |
| | Regulatory and Compliance Changes | Privacy regulations limit the use of certain features critical for predictions [83] |
| | Adversarial Attacks and Data Poisoning | Falsifying patient records skews model predictions [84] |
| | Ethical Constraints and Bias Mitigation Efforts | Bias-mitigation interventions can change which features are used for prediction [7] |
| | Changes in Measurement Techniques | New lab assay for troponin provides results that aren't comparable to the previous method [85] |
| | Changes in Population Health Trends | Aging populations or increases in chronic conditions shift the input feature distribution [81] |
| **Label S** | Seasonal Disease Prevalence | Flu prevalence increases in winter, changing the proportion of positive cases [86] |
| | Seasonal Health Campaigns | Public health campaigns can influence patient behavior [81] |
| | Weather-related health issues | Weather-related health issues can arise due to heatwaves or cold snaps [81] |
| **Concept Drift** | Changes in Treatment Protocols | New clinical guidelines alter the relationship between symptoms and outcomes [81] |
| | Feedback Loops | Additional care prevents readmission, and the model misinterprets this as reduced risk [87] |
| | External Changes in Clinical Practice | Adoption of new surgical techniques changes complication patterns [81] |
| | Changes in Related Policies or Economic Factors | Insurance policy changes alter patient behavior and readmission patterns [81] |
| | Emergence of Unmeasured Confounding Variables | Environmental hazards change disease presentation patterns [88] |
| | Interaction Effects from Concurrent Models | A new AI triage system changes patient flow and case severity [89] |
| | Evolution of Disease Characteristics | New variants alter symptom-outcome relationships [90] |

Table 2: Reasons and examples for covariate shift, label shift and concept drift in the healthcare domain.

# B  Label Shift

Following the taxonomy in [26], label shift (also called prior probability shift or target shift) corresponds to the case where the marginal distribution of outcomes changes while the conditional distribution of features given outcomes remains fixed. Formally, label shift occurs when $p_{t_1}(y) \neq p_{t_0}(y)$ for some $y \in \mathcal{Y}$, while $p_{t_1}(\mathbf{s}, \mathbf{c} \mid y) = p_{t_0}(\mathbf{s}, \mathbf{c} \mid y)$ holds $\forall y \in \mathcal{Y}$ and $\forall (\mathbf{s}, \mathbf{c}) \in \mathcal{S} \times \mathcal{C}$. This type of shift is prevalent in the settings, where disease prevalence naturally fluctuates due to seasonal patterns, public health interventions, demographic changes, and evolving clinical practices.

To detect label shift, we formulate a hypothesis test on the marginal label distribution. Let $p_{t_0}(y)$ and $p_{t_1}(y)$ denote the marginal distributions of outcomes at $t_0$ and $t_1$. We test:

$$
\begin{aligned}
H_0 : \quad & p_{t_0}(y) = p_{t_1}(y) \quad \forall y \in \mathcal{Y} \\
H_1 : \quad & p_{t_0}(y) \neq p_{t_1}(y) \quad \exists y \in \mathcal{Y}
\end{aligned}
\tag{10}
$$

Note: we assume that $p_{t_1}(\mathbf{s}, \mathbf{c} \mid y) = p_{t_0}(\mathbf{s}, \mathbf{c} \mid y)$ holds $\forall y \in \mathcal{Y}$ and $\forall (\mathbf{s}, \mathbf{c}) \in \mathcal{S} \times \mathcal{C}$. When label shift is detected, several mitigation strategies are available that do not require full model retraining [91]. For probabilistic classifiers, output probabilities can be adjusted using the ratio $\frac{p_{t_1}(y)}{p_{t_0}(y)}$ to account for changed base rates, while binary classifiers can undergo decision threshold recalibration to maintain desired operating characteristics (e.g., sensitivity/specificity trade-offs) under the new class distribution. When retraining is feasible, importance weights $w_i = \frac{p_{t_1}(y_i)}{p_{t_0}(y_i)}$ can be applied to training samples to simulate the target distribution [43]. Additionally, Expectation-Maximization approaches such as Black Box Shift Estimation (BBSE) [43] can estimate label shift and correct predictions using only unlabeled post-deployment data and a confusion matrix, though these methods carry their own assumptions and limitations.

**OP: Label Scarcity**  Detecting label shift requires access to ground truth labels $Y$ at both pre- and post-deployment stages to estimate $p_{t_0}(y)$ and $p_{t_1}(y)$. While label shift detection requires fewer labels than concept drift detection (only marginal distributions rather than joint distributions), the fundamental label scarcity problem remains. Active sampling strategies that prioritize diverse samples across the feature space (rather than focusing on model uncertainty as in active learning) may be more appropriate for label shift detection. Stratified sampling approaches that ensure representation across key demographic and clinical subgroups can improve prevalence estimation efficiency.

# C  Performance Metrics

Selecting appropriate metrics is critical to assessing model degradation. Binary classification tasks in healthcare require complementary metrics that capture different aspects of clinical performance and align with specific medical decision-making needs. While accuracy provides an overall measure of correctness, sensitivity, and specificity, offer insights into a model's ability to identify positive and negative cases, respectively - particularly important when false negatives (missed diagnoses) or false positives (unnecessary interventions) carry different clinical consequences. Table 3 provides a comprehensive overview of these metrics, their mathematical formulations, and their distribution assumptions. Understanding these properties, particularly their asymptotic behavior and required assumptions, is essential for constructing valid statistical tests for performance degradation.

Accuracy measures the overall proportion of correct predictions but can be misleading in healthcare settings where class imbalance is common. For instance, a model predicting a rare disease might achieve high accuracy by simply predicting "negative" for all cases. Precision quantifies how often positive predictions are correct, which is crucial in scenarios where false positives lead to unnecessary interventions. Recall (sensitivity) measures the model's ability to identify actual positive cases, vital in conditions where missing a diagnosis could be life-threatening. Conversely, specificity indicates the model's ability to correctly identify negative cases, particularly when false positives could lead to unnecessary, expensive, or risky procedures.

Positive and Negative Predictive Values (PPV and NPV) are particularly relevant for clinical decision-making as they answer the physician's primary question: given the model's prediction, what is the probability it is correct? The F1 score balances precision and recall, useful when false positives and

| Name | Metric | Distribution Assumptions | Distribution |
|---|---|---|---|
| **Accuracy** | $\dfrac{\text{TP} + \text{TN}}{\text{TP} + \text{FP} + \text{FN} + \text{TN}}$ | - Observations are independent.
- Number of correct predictions follows a binomial distribution.
- Sample size large enough for normal approximation $(np \geq 5),\, n(1-p) \geq 5)$. | Binomial distribution |
| **Precision (Positive Predictive Value (PPV))** | $\dfrac{\text{TP}}{\text{TP} + \text{FP}}$ | - Observations are independent.
- Number of true positives among predicted positives follows a binomial distribution.
- Large sample size for normal approximation $(np \geq 5),\, n(1-p) \geq 5)$. | Binomial distribution |
| **Recall (Sensitivity)** | $\dfrac{\text{TP}}{\text{TP} + \text{FN}}$ | - Observations are independent.
- Number of true positives among actual positives follows a binomial distribution.
- Large number of positive cases for normal approximation. | Binomial distribution |
| **F1 Score** | $2 \times \dfrac{\text{Precision} \times \text{Recall}}{\text{Precision} + \text{Recall}}$ | - Complex function of two proportions (Precision and Recall).
- Distribution is not easily defined analytically.
- Normal approximation may not be appropriate even with large $n$
- bootstrap methods recommended for inference $n$ | Null distribution is unknown |
| **Specificity** | $\dfrac{\text{TN}}{\text{TN} + \text{FP}}$ | - Observations are independent.
- Number of true negatives among actual negatives follows a binomial distribution.
- Large number of negative cases for normal approximation. | Binomial distribution |
| **Negative Predictive Value (NPV)** | $\dfrac{\text{TN}}{\text{TN} + \text{FN}}$ | - Observations are independent.
- Number of true negatives among predicted negatives follows a binomial distribution.
- Large sample size for normal approximation. | Binomial distribution |
| **Balanced Accuracy** | $\dfrac{\text{Sensitivity} + \text{Specificity}}{2}$ | - Average of two proportions (Sensitivity and Specificity).
- Assumes independence between Sensitivity and Specificity estimates.
- Normal approximation may be used if both components have normal distributions.
- Approximate normal distribution under large $n$ | Exact distribution is complex |

Table 3: Distribution Assumptions for Common Performance Metrics

negatives have significant clinical implications. Balanced accuracy, the average of sensitivity and specificity, provides a more representative performance measure for imbalanced datasets, common in medical conditions with low prevalence.

Most performance metrics mentioned in this section follow binomial distributions, reflecting their foundation in counting correct and incorrect predictions. Under sufficient sample sizes specified in Table 3, these metrics converge to normal distributions through the Central Limit Theorem. This convergence occurs when we have enough samples of each class ($np \geq 5$ and $n(1-p) \geq 5$) for fundamental metrics like accuracy and precision. The resulting normal approximation enables straightforward statistical inference through confidence intervals and hypothesis tests. However, composite metrics require more careful statistical consideration. The balanced accuracy, while still asymptotically normal, has a variance that must account for the relationship between its components. The F1 score presents even greater challenges due to its nonlinear nature as a harmonic mean of precision and recall. Its sampling distribution resists simple analytical characterization, necessitating bootstrap methods or the delta method for reliable inference in practice.

## D    Choosing A Statistical Hypothesis Test and Heuristics

Statistical hypothesis testing relies on knowing the distribution of the test statistic under the null hypothesis. If this null distribution is known, we can directly compute the probability of observing a given test statistic and define a corresponding critical region. However, in many practical settings, the null distribution is not known and must be estimated. One approach is to assume a specific parametric form (e.g., Gaussian) and estimate its parameters from data. Alternatively, non-parametric methods such as permutation tests [92, 93] make fewer assumptions and instead rely on data-driven resampling procedures. We broadly categorize tests into *parametric* and *non-parametric*, as summarized in Table 1.

In addition to formal test statistics, our framework includes heuristics—such as control charts, process monitoring techniques, and distance or divergence measures—as practical tools for detecting distribution shifts. It is essential to distinguish between statistically rigorous hypothesis tests, including both parametric and non-parametric approaches, and heuristic methods. Parametric tests, such as the t-test or F-test, rely on assumptions like normality and independence and typically require larger sample sizes; however, they offer precise Type I and Type II error control under these conditions. Non-parametric tests, such as the Kolmogorov–Smirnov test or the Friedman–Rafsky

test, are equally valid in a statistical sense and provide robust inference without strong distributional assumptions. In contrast, heuristics like the Wasserstein Distance or Maximum Mean Discrepancy can be powerful in practice—especially in high-dimensional or complex settings—but often rely on empirically determined thresholds and lack formal guarantees on error rates. Table 1 summarizes these methods, and their detailed mathematical formulations are provided in Appendices F and G.

Selecting an appropriate evaluation method for detecting shifts in clinical data distributions requires careful consideration of the type of change, the nature of the data, and the practical constraints in post-deployment settings. Table 1 summarizes the range of test statistics and heuristics available for detecting changes in means, variances, or full distributions. Below, we provide a guided walkthrough of how to select among these tools, grounded in real-world examples and tradeoffs inherent to each method.

**Determine parametric vs. non-parametric regime.**   The first key decision is whether to adopt a parametric or non-parametric approach. Parametric tests assume data follow a known distribution—typically Gaussian—and offer efficient, high-power tests when these assumptions hold. However, real-world post-deployment data often violate these assumptions: distributions may be skewed, heavy-tailed, or multi-modal; features may be categorical, ordinal, or continuous; and high-dimensional settings are common (e.g., embeddings, imaging, multimodal EHRs). In such cases, non-parametric methods are more robust.

**Define the type of shift of interest.**   If the goal is to detect changes in the mean of a feature or model output, parametric methods like the *z-test* (requires known variance and normality) or the *two-sample t-test* (assumes equal variance) are natural starting points. When variance equality is uncertain, *Welch's t-test* relaxes that assumption and provides robust inference under heteroskedasticity. For instance, a shift in average glucose levels between pre- and post-deployment periods can be assessed using these tests. For variance shifts, the *F-test* compares two variances under normality, while *Bartlett's test* generalizes this to multiple groups with better stability. However, both are sensitive to non-normality. In such cases, *Levene's test*, a non-parametric alternative, offers robustness at the cost of slightly lower power.

**Select tests suited to your feature space and dimensionality.**   When monitoring high-dimensional, heterogeneous, or structured data—such as EHR records, where inputs include demographics, vitals, and lab values—methods like *Energy Distance* and *Maximum Mean Discrepancy (MMD)* are advantageous. MMD, in particular, is effective in detecting subtle distributional changes in image embeddings or textual representations, assuming an appropriate kernel is chosen. Similarly, *Wasserstein Distance*, rooted in optimal transport theory, captures support and shape shifts (e.g., population drift) even when distributions do not overlap. For distribution pairs with full support overlap, *Kullback-Leibler (KL) divergence* or its symmetric counterpart, *Jensen-Shannon (JS) divergence*, are informative but require density estimation, which may be infeasible in high dimensions.

**Consider temporal monitoring and real-time detection.**   For settings that require ongoing monitoring of streaming features (e.g., tracking patient inflow distributions or model prediction confidence), univariate process control tools offer lightweight yet powerful diagnostics. *Shewhart control charts* are designed to detect sudden shifts in feature means (e.g., a sudden increase in patient temperature), while *CUSUM charts* accumulate deviations over time to detect persistent small changes. *EWMA (Exponentially Weighted Moving Average)* charts offer smoother detection of gradual changes and are particularly useful when the underlying process drifts slowly, as may occur with seasonal disease incidence or chronic care trends. While these methods assume univariate i.i.d. data, they can be extended to multivariate settings using multivariate statistical process control (MSPC) methods, albeit with stronger distributional assumptions.

**Practical limitations and methodological tradeoffs.**   Each method comes with tradeoffs. Parametric tests like *t-* and *F-tests* are statistically efficient but brittle under assumption violations. Non-parametric tests (e.g., *KS*, *MMD*, *Wasserstein Distance*) are flexible but often require larger samples for power, careful kernel or metric selection, and suffer from the "curse of dimensionality." Tests like *Friedman–Rafsky*, which uses graph-based minimum spanning tree construction, are especially useful for multivariate shifts but can be computationally intensive. In addition, tests such as *JS Divergence* or *Energy Distance* may be hard to interpret clinically without well-defined thresholds.

Therefore, practitioners must balance statistical power, interpretability, computational burden, and alignment with clinical relevance when choosing an evaluation method.

**In summary** There is no universally optimal method for detecting distributional shifts. Instead, Table 1 provides a toolbox for context-specific decision making. When assumptions are met, parametric tests offer high power and clean interpretability. For complex, high-dimensional, or weakly labeled post-deployment settings, non-parametric distributional distances—such as *MMD*, *Wasserstein*, and *Friedman–Rafsky*—are more robust and generalizable. Refer to Appendices F and G for the descriptions of the parametric and non-parametric tests, respectively.

# E Turning Heuristics into Statistical Tests

| | | Heuristics | When to Use / Notes | Data Distribution Assumptions |
|---|---|---|---|---|
| **Param.** | **M** | Shewhart Control Charts | For process monitoring; detects sudden shifts | normality, stable process, markov process |
| | | CUSUM Chart | For detecting small, persistent shifts over time | stable process, known target value |
| | | EWMA | For detecting gradual changes with weighted historical data | stable baseline, mean stationarity |
| **Non-Param.** | **Distr. Shifts** | Energy Distance | Measures statistical distances between distributions | |
| | | Wasserstein Distance | When distributions have little or no overlap; captures shape/support shifts | |
| | | Kullback-Leibler (KL) Divergence | When distributions have complete support overlap; information-theoretic interpretation; asymmetric measure | |
| | | Jensen-Shannon (JS) Divergence | Bounded symmetric variant of KL divergence; use if distributions may not overlap; symmetric measure | |

Table 4: Summary of two sample test statistics and heuristics for detecting differences between $p_{t_0}$ and $p_{t_1}$, including assumptions and use cases. All methods assume i.i.d. data. Note: "M" denotes mean, "V" denotes variance

While divergence measures such as *Jensen-Shannon divergence*, *Maximum Mean Discrepancy (MMD)*, *Energy Distance*, and *Wasserstein distance* provide powerful tools for quantifying dissimilarity between distributions, they are not hypothesis tests on their own. To formally test whether two distributions differ, these measures must be embedded within a hypothesis testing framework that controls Type I and Type II errors. We now describe a general procedure to transform any such divergence into a valid two-sample test using permutation testing [92, 93].

Let $D(\mathcal{D}_{t_0}, \mathcal{D}_{t_1})$ be any divergence or distance-based dissimilarity measure between distributions, such as: Jensen-Shannon divergence, Energy Distance or Wasserstein Distance.

**Permutation-Based Hypothesis Testing Procedure** The permutation test simulates the distribution of a test statistic (e.g., Jensen–Shannon divergence) under the null hypothesis $H_0$, and computes the $p$-value using this null distribution based on the observed two samples, $\mathcal{D}_{t_0}$ and $\mathcal{D}_{t_1}$, to determine whether to reject $H_0$. The permutation test is performed as follows.

1. **Compute Observed Statistic:**
$$T_{\text{obs}} = D(\mathcal{D}_{t_0}, \mathcal{D}_{t_1})$$

2. **Construct the Null Distribution via Permutation:**
   - Pool the data: $\mathcal{D} = \mathcal{D}_{t_0} \cup \mathcal{D}_{t_1}$
   - For $B$ iterations (e.g., $B = 1000$):
     (a) Randomly permute the labels of the pooled dataset.
     (b) Split the permuted data into two groups of sizes $n_0$ and $n_1$ according to the permuted labels.
     (c) Compute the test statistic $T_b$ using (1) based on the two permuted groups.
   - This yields an empirical null distribution $\{T_1, \ldots, T_B\}$.

3. **Compute the $p$-value (right-sided)[5]:**
$$p = \frac{1}{B} \sum_{b=1}^{B} \mathbb{I}\left(T_b \geq T_{\text{obs}}\right).$$

4. **Make a Decision:** Reject $H_0$ if $p < \alpha$, for a chosen significance level $\alpha$ (e.g., 0.05).

---

[5]Left-sided or two-sided $p$-values can be computed analogously without loss of generality.

**Benefits and Limitations** This approach makes minimal assumptions—it is *non-parametric*, applicable in *high-dimensional* settings, and works with *mixed or complex data types.* However, its statistical power depends on the choice of divergence measure, the sample size, and the number of permutations $B$. Care must also be taken when the divergence relies on kernel or transport parameters, which should be selected independently of the test data to avoid selection bias.

# F  Parametric Tests and Heuristics

## F.1  Mean Shift

**z test [94]** is a parametric test used to determine whether the means of two independent populations differ significantly, under the assumption that population variances are known. The test statistic is:

$$z = \frac{\bar{X}_0 - \bar{X}_1}{\sqrt{\frac{\sigma_0^2}{n_0} + \frac{\sigma_1^2}{n_1}}}$$

where $\bar{X}_0$ and $\bar{X}_1$ are the sample means from the pre- and post-deployment periods, and $\sigma_0^2, \sigma_1^2$ are the known variances of the metric in each period. Under the null hypothesis, $Z \sim \mathcal{N}(0, 1)$, and a two-sided p-value can be computed accordingly. We reject the null hypothesis if:

$$|z| > z_{\alpha/2}$$

where $z_{\alpha/2}$ is the critical value from the standard normal distribution (mean 0, variance 1). In practice, the z-test is appropriate when the sample sizes are large (invoking the Central Limit Theorem) or when the variances are reliably estimated from historical data. Despite its simplicity, it provides a strong baseline for detecting statistically significant changes in model performance.

**Two-Sample t-Test [95]** assesses whether the means of two independent samples differ significantly, assuming normally distributed data with equal variances. The test statistic is given by:

$$t = \frac{\bar{X}_0 - \bar{X}_1}{s_p \cdot \sqrt{\frac{1}{n_0} + \frac{1}{n_1}}}$$

$$s_p = \sqrt{\frac{(n_0 - 1)s_0^2 + (n_1 - 1)s_1^2}{df}}$$

where $\bar{X}_k$ and $s_k^2$ denote the sample mean and variance of group $k$, and $n_k$ represents the sample size, with degrees of freedom $df = n_0 + n_1 - 2$. The null hypothesis ($H_0$) assumes no difference in means ($\mu_0 = \mu_1$), while the alternative hypothesis ($H_1$) suggests a shift in mean. This test is appropriate for detecting *mean shifts* in $p_{t_1}$ (the post-deployment distribution) when normality assumptions hold. With the critical value of

$$|t| > t_{\alpha/2,df}$$

where $t_{\alpha/2,df}$ is the critical value from Student's t-distribution.

**Welch's t-Test [96]** When variances are unequal, Welch's t-test modifies the degrees of freedom using the Welch-Satterthwaite equation, improving robustness. Welch's t-test is used to compare the means of two samples when the assumption of equal variances does not hold.

$$t = \frac{\bar{X}_0 - \bar{X}_1}{\sqrt{\frac{s_0^2}{n_0} + \frac{s_1^2}{n_1}}}$$

To test the null hypothesis $H_0 : \mu_1 = \mu_2$, we calculate the critical value $t_{\alpha/2,df}$ from the t-distribution with $df$ degrees of freedom. Reject $H_0$ if:

$$|t| > t_{\alpha/2, df}.$$

The degrees of freedom are computed using the Welch-Satterthwaite equation:

$$df = \frac{\left(\frac{s_0^2}{n_0} + \frac{s_1^2}{n_1}\right)^2}{\frac{\left(\frac{s_0^2}{n_0}\right)^2}{n_0 - 1} + \frac{\left(\frac{s_1^2}{n_1}\right)^2}{n_1 - 1}}$$

**Shewhart Control Charts [97]** track the mean ($\mu$) and standard deviation ($\sigma$) of continuous variables, in our case one of the features of the model, flagging when values by examining if the newly collected values are outside the upper control limit (UCL) and lower control limit (LCL):

$$\text{UCL} = \mu_0 + L\sigma_0, \quad \text{LCL} = \mu_0 - L\sigma_0$$

where $\mu_0$, $\sigma_0$ are baseline parameters, in our case values during the pre-deployment period, and $L$ (typically 3 for 99.73% confidence) sets control limits. Despite detecting large shifts, this method presents a limitation as it only uses a single point in time for evaluation and does not consider the dynamics of change. Small shifts in the distribution may go undetected.

**Cumulative Sum (CUSUM) Charts [98]** addresses the limitations, by considering the current and historical values, accumulating deviations from target values to detect small but persistent shifts in feature distributions. CUSUM Charts accumulate deviations from the target value $\mu_0$:

$$S_t^+ = \max(0, S_{t-1}^+ + (x_t - \mu_0) - k)$$
$$S_t^- = \min(0, S_{t-1}^- + (-x_t + \mu_0) - k)$$

where $x_t$ is the value of the feature at time $t$, $k$ is a reference value chosen for detection sensitivity and $S_0^+ = S_0^- = 0$. Given a control limit $h > 0$ is, the decision rule is defined by

$$S_t^+ > h$$
$$S_t^- < -h$$

While CUSUM Charts achieve the desired goal while addressing the limitations, there are a few drawbacks. Once the shift is detected, the next detection process has to restart from the initial value.

**EWMA (Exponentially Weighted Moving Average) Charts [99]** provide more convenience without jeopardizing the performance of CUSUM charts. The chart calculates the weighted average of the historical data up to the current time; by weighting recent observations more heavily to identify emerging trends. EWMA Charts are defined as:

$$E_t = \lambda x_t + (1 - \lambda)E_{t-1}$$

Where $E_0 = \mu_0$, $\lambda \in (0, 1]$ is the weighting parameter. The control limits are given by:

$$\sigma_{E_t}^2 = \frac{\lambda}{2 - \lambda}(1 - (1 - \lambda)^{2t})\sigma_0^2$$
$$\text{UCL}_t = \mu_0 + \rho\sigma_{E_t}$$
$$\text{LCL}_t = \mu_0 - \rho\sigma_{E_t}$$

where $\rho$ is a parameter. Note: we assume that variance remains unchanged after the distribution shift.

The univariate approaches discussed above are computationally efficient, but they can miss complex feature interactions and face multiple testing challenges when monitoring many features simultaneously [100]. While all of the presented methods can be generalized to the multivariate setting, known as Multivariate Statistical Process (MSPC) control charts. The major limitation is the assumption that the processes distribution is multivariate normal [101], methods may not capture complex nonlinear relationships.

## F.2 Variance Shift

**F-Test [102]** is used to compare the variances of two independent samples to determine if they are significantly different, assuming normal distribution. It is based on the ratio of sample variances:

$$F = \frac{s_0^2}{s_1^2}$$

where $s_0^2$ and $s_1^2$ are the sample variances of groups 0 and 1. The null hypothesis assumes equal variances ($H_0 : \sigma_0^2 = \sigma_1^2$). The test statistic follows an F-distribution with degrees of freedom: $df_1 = n_0 - 1$, $df_2 = n_1 - 1$. The critical value is obtained from the F-distribution table at the chosen significance level $\alpha$, denoted as $F_{\alpha, df_1, df_2}$. The critical value is:

$$F_{critical} = F_{\alpha, df_1, df_2}.$$

**Bartlett's Test [103]** Bartlett's test assesses whether multiple groups have equal variance under the assumption of normality. It is more sensitive to deviations from normality than Levene's test. The test statistic, for 2 distributions is:

$$B = \frac{(n_0 + n_1 - 2) \ln s_p^2 - \sum_{i=0}^{1} (n_i - 1) \ln s_i^2}{1 + \frac{1}{3} \left( \sum_{i=0}^{1} \frac{1}{n_i - 1} - \frac{1}{n_0 + n_1 - 2} \right)}$$

$$s_p^2 = \frac{(n_0 - 1)s_0^2 + (n_1 - 1)s_1^2}{n_0 + n_1 - 2}$$

where: $s_i^2$ is the sample variance of group, $s_p^2$ is the pooled variance:

The test statistic follows a chi-square distribution with $df = 1$, with the critical value of:

$$B_{critical} = \chi_{\alpha, 1}^2.$$

# G Non-Parametric Tests and Heuristics

## G.1 Mean Shift

Evaluating the hypothesis test requires methods that can detect distributional changes across different scales. For individual features, classical statistical approaches provide efficient monitoring of univariate distributions. For the full joint distribution, distance and divergence measures enable direct hypothesis testing in high-dimensional spaces. Relying on the methodology described in [101], we outline the Statistical Process Control (SPC) methods test for shifts in univariate or low-dimensional projections of the data.

**Mann-Whitney U Test [104]** The Mann-Whitney U test (Wilcoxon rank-sum test) is a non-parametric alternative for comparing median shifts between two samples:

$$U = n_0 n_1 + \frac{n_0(n_0 + 1)}{2} - R_0$$

where $R_0$ is the sum of ranks in sample 0. The critical value is obtained from standard U-statistic tables (e.g. for $n_0 = n_1 = 20$ and $\alpha = 0.05$ $c = 127$).

## G.2 Varience Shift

**Levene's Test [105]** Levene's test is a robust alternative to the F-test for comparing variances when normality cannot be assumed. It tests whether multiple groups have equal variance by transforming data into deviations from the group mean or median. The test statistic for 2 distributions is:

$$W = (n_0 + n_1 - 2) \cdot \frac{\sum_{i=0}^{1} n_i (Z_{i.} - Z_{..})^2}{\sum_{i=0}^{1} \sum_{j=1}^{n_i} (Z_{ij} - Z_{i.})^2}$$

where: $N$ is the total number of observations, $k$ is the number of groups, $Z_{ij} = |X_{ij} - \bar{X}_i|$ (absolute deviations from group means or medians), $Z_{i.} = \frac{1}{n_i} \sum_{j=1}^{n_i} Z_{ij}$, $Z_{..} = \frac{1}{N} \sum_{i=0}^{1} \sum_{j=1}^{n_i} Z_{ij}$. The test statistic follows an F-distribution with $df_1 = k - 1$ and $df_2 = N - k$. The critical value is:

$$W_{critical} = F_{\alpha, 1, n_0 + n_1 - 2}.$$

## G.3 Distribution Shift

**Kolmogorov-Smirnov (KS) Test [106, 107]**  The KS test evaluates differences between empirical cumulative distribution functions (ECDFs):

$$D = \sup_x |F_0(x) - F_1(x)|$$

$$D_{critical} = c(\alpha) \sqrt{\frac{n_0 + n_1}{n_0 n_1}}$$

where $c(\alpha)$ is a constant based on the significance level (e.g. 1.36 for $\alpha = 0.05$)

**Anderson-Darling Test [108]**  evaluates whether a sample follows a given distribution, improving upon the Kolmogorov-Smirnov test by giving more weight to the tails. For a sample of size $n$, the test statistic is:

Let $X_0$ and $X_1$ be pre-deployment and post-deployment samples of sizes $n_0$ and $n_1$, respectively, with $N = n_0 + n_1$. Denote by $Z_{(1)}, \ldots, Z_{(N)}$ the pooled and ordered combined sample, and let $H_j$ be the number of observations from $X_0$ among $\{Z_{(1)}, \ldots, Z_{(j)}\}$. The two-sample Anderson–Darling test statistic is defined as:

$$A_{n_0 n_1}^2 = \frac{1}{n_0 n_1} \sum_{j=1}^{N-1} \frac{(H_j N - n_0 j)^2}{j(N-j)}.$$

Critical values for the Anderson-Darling test depend on the distribution being tested. For a normal distribution, significance thresholds are tabulated, with rejection occurring if:

$$A_{critical}^2 = A_\alpha^2$$

**The Friedman-Rafsky test [109]**  is a multivariate nonparametric, graph-based test used to determine whether two samples are drawn from the same distribution. Given two combined samples $Z$ from $p_{t_0}$ and $p_{t_1}$, The Friedman-Rafsky test constructs the Minimum Spanning Tree (MST) $T$ over $Z$, where each point in $Z$ is a node and edges are weighted by the distance $d(x, y)$ (e.g., Euclidean distance) between points. The test statistic is the number of edges in the MST that connect points from different groups (cross-edges). Let $R$ denote the total number of runs (or clusters) in the MST, where a "run" is defined as a sequence of connected nodes belonging to the same group (either $X$ or $Y$).

$$R = \sum_{(i,j) \in T} \mathbb{1}_{\text{group}(x_i) \neq \text{group}(x_j)}$$

Note $R$ counts the number of edges connecting points from different samples.

The critical region for rejection of $H_0$ is determined via permutation testing, where the labels of $X$ and $Y$ are randomly permuted to generate the null distribution of $R$. Reject $H_0$ if standardized

statistic exceeds critical value:

$$\frac{R - \mathbb{E}[R]}{\sqrt{\text{Var}(R)}} > c_\alpha$$

where:

$$\mathbb{E}[R] = \frac{2n_0 n_1}{n_0 + n_1 - 1}$$
$$\text{Var}(R) = \frac{2n_0 n_1}{(n_0 + n_1)(n_0 + n_1 - 1)} \left( 1 + \frac{W - (n_0 + n_1 - 1)}{2(n_0 + n_1 - 2)} \right)$$

where $n_0, n_1$ are sample sizes and $W$ is the number of cross-edges in pairs of adjacent edges in $T$.

**Energy Distance [110]** is a nonparametric measure of the distance between two probability distributions $P$ and $Q$. It is derived from the concept of statistical potential energy, where the "energy" depends on pairwise distances between points in the distributions. The Energy Distance is directly related to the distance between characteristic functions of the two distributions and can be used to conduct two-sample tests. This metric does not require density estimation and is particularly useful for comparing high-dimensional or non-Euclidean distributions. The Energy Distance is defined as:

$$D_E(p_{t_0}, p_{t_1}) = 2\mathbb{E}[d(X, Y)] - \mathbb{E}[d(X, X')] - \mathbb{E}[d(Y, Y')],$$

where $d$ is the distance metric (e.g., Euclidean distance) and $X, X' \sim p_{t_0}, Y, Y' \sim p_{t_1}$.

**Maximum Mean Discrepancy (MMD) [79]** measures distribution distances in reproducing kernel Hilbert space, avoiding explicit density estimation by comparing statistical moments of the distributions. The MMD between two probability distributions at two different times $p_{t_0}$ and $p_{t_1}$ is defined as:

$$\text{MMD}(p_{t_0}, p_{t_1}) = \left\| \mathbb{E}_{X \sim p_{t_0}}[\varphi(X)] - \mathbb{E}_{Y \sim p_{t_1}}[\varphi(Y)] \right\|_{\mathcal{H}}$$

where $\varphi : \mathcal{X} \to \mathcal{H}$ is a feature mapping function that maps elements from the input space $\mathcal{X}$ to a reproducing kernel Hilbert space (RKHS) $\mathcal{H}$. For example, a Gaussian or Laplacian kernel, $k(x, \cdot)$. To test the statistical significance using empirical data we can calculate Biased Empirical Estimate of MMD ($\text{MMD}_b$) and employ the following acceptance region:

$$\text{MMD}_b(p_{t_0}, p_{t_1}) = \left[ \frac{1}{m^2} \sum_{i,j=1}^{m} k(x_i, x_j) - \frac{2}{mn} \sum_{i=1}^{m} \sum_{j=1}^{n} k(x_i, y_j) + \frac{1}{n^2} \sum_{i,j=1}^{n} k(y_i, y_j) \right]^{\frac{1}{2}}$$
$$\text{MMD}_b(p_{t_0}, p_{t_1}) < \sqrt{\frac{2K}{m}} (1 + \sqrt{2 \log \alpha^{-1}})$$

where, $\alpha$ is the hypothesis test level (e.g., 0.05), $K$ is the upper bound on the kernel function (1 for a normalized kernel), and $m$, $n$ are sample sizes from each distributions.

High-Dimensional Distribution Testing divergence measures enable comprehensive hypothesis testing in high-dimensional spaces. These methods build on statistical divergence estimation [111] and kernel methods [31].

**Wasserstein distance [112]** motivated by the optimal transport theory [113] provides theoretically grounded distribution comparisons [27] by measuring the minimum "cost" of transforming one distribution into another. These distances are especially useful in healthcare applications as they account for the underlying geometry of the feature space. Recent advances in computational optimal transport have made these methods practical for high-dimensional medical data. The Wasserstein

distance of order $p$ between two probability distributions at different points in time $p_{t_0}$ and $p_{t_1}$ on a metric space $(\mathcal{X}, d)$ is defined as:

$$W_p(p_{t_0}, p_{t_1}) = \left( \inf_{\pi \in \Pi(p_{t_0}, p_{t_1})} \int_{\mathcal{X} \times \mathcal{X}} d(x, y)^p \, d\pi(x, y) \right)^{1/p}$$

where $d(x, y)$ is the metric (or distance function) on the space $\mathcal{X}$. $\Pi(P, Q)$ is the set of all joint probability distributions (also called couplings) $\pi(x, y)$ on $\mathcal{X} \times \mathcal{X}$ such that the marginal distributions are $p_{t_0}$ and $p_{t_1}$, i.e., $\int_{\mathcal{X}} \pi(x, y) \, dx = p_{t_1}(y), \quad \int_{\mathcal{X}} \pi(x, y) \, dy = p_{t_0}(x)$.

**The family of f-Divergences [114]**, including Kullback-Leibler (KL) divergence [115] and Jensen-Shannon divergence [116], offer another approach to distribution comparison. While these methods provide strong theoretical guarantees, they require density estimation which can be challenging in high dimensions. The (KL) divergence between two probability distributions $p_{t_0}$ and $p_{t_1}$ over a shared domain $\mathcal{X}$ is defined as:

$$D_{\mathrm{KL}}(p_{t_0} \| p_{t_1}) = \int_{\mathcal{X}} p_{t_0}(x) \log \frac{p_{t_0}(x)}{p_{t_1}(x)} \, dx,$$

provided that $p_{t_0}(x) > 0 \implies p_{t_1}(x) > 0$ for all $x \in \mathcal{X}$.

The JS divergence between two probability distributions $p_{t_0}$ and $p_{t_1}$ is a symmetric and bounded measure defined as:

$$D_{\mathrm{JS}}(p_{t_0} \| p_{t_1}) = \frac{1}{2} D_{\mathrm{KL}}(p_{t_0} \| M) + \frac{1}{2} D_{\mathrm{KL}}(p_{t_1} \| M)$$

where $M = \frac{1}{2}(p_{t_0} + p_{t_1})$ is the average distribution.

