# OpenReview forum: "Statistically Valid Post-Deployment Monitoring Should Be Standard for AI-Based Digital Health"
_NeurIPS.cc/2025/Position_Paper_Track — NeurIPS 2025 Position Paper Track_

### Official Review · Reviewer_8KGL · 2025-07-25

**Significance:** 3
**Presentation:** 4
**Rating:** 7
**Confidence:** 4

**Summary:**

The paper argues that post-deployment monitoring of clinical AI systems is critically underdeveloped and advocates for a shift toward statistically valid, label-efficient hypothesis testing frameworks. It highlights that despite clinical AI tools’ susceptibility to performance degradation due to factors like covariate shift and concept drift, only 9% of FDA-registered models include surveillance plans. The authors propose formalizing monitoring as two-sample hypothesis tests, encompassing both data shift detection and performance monitoring, with rigorous statistical guarantees. They present detailed formulations for detecting changes in distribution and performance, introduce open challenges such as label scarcity and subgroup identification, and contrast their approach against alternatives like continual learning, Bayesian change-point detection, and conformal methods, ultimately positioning hypothesis testing as the most robust, interpretable, and regulator-aligned method.

**Strengths:**

* The paper is well-motivated, grounded in regulatory context, and presents a principled, formal approach to a neglected yet critical problem in clinical AI. Its framing of monitoring as hypothesis testing is both rigorous and elegant, offering statistical guarantees often missing in heuristic-based MLOps
* The treatment of label scarcity, through surrogate labeling and active learning, adds practical depth, and the discussion of subgroup-specific degradation addresses fairness and transparency
* I think the paper is well written all around

**Weaknesses:**

* the paper leans heavily on assumptions like i.i.d. sampling and known pre-deployment distributions, which may not hold in real-world hospital systems
* the authors acknowledge the high dimensionality of clinical data, practical strategies for selecting and validating appropriate test statistics in such settings are underexplored. The reliance on statistical tests may also struggle with the complexity and multimodality of clinical feature spaces, especially when concept drift and covariate shift co-occur.

**Questions:**

* How would this framework generalize to multi-modal clinical data, such as image-plus-text inputs, where joint distributions are even more complex? I ask this because your title contains Digital Health but multimodality is not considered in the works.
* how does the framework handle confounding variables or unobserved shifts in label semantics

**Alternative Position:**

Yes, and alternative positions are well-considered and named but not addressed

**Author Identification:**

No.

**Context:**

3

**Discussion:**

3

**Ethics:**

["NO or VERY MINOR ethics concerns only"]

**Position:**

Yes, the paper argues for or against a position related to machine learning.

**Support:**

3

**Thoroughness:**

4

---

### Official Review · Reviewer_ZCZH · 2025-08-05

**Significance:** 4
**Presentation:** 3
**Rating:** 7
**Confidence:** 4

**Summary:**

The manuscript argues for statistically grounded post-deployment validation of medical devices. Two types performance degradations are distinguished:

(1) covariate shift - changes in input patterns
(2) concept drift - changes in relationships between features and labels

Accurate and timely “catching” of these errors will prevent patient harm associated with incorrect AI decision-making.

In particular, the manuscript highlights the value of detecting performance degradations using hypothesis testing, and discusses the pros and cons of the proposed method compared to other techniques. Finally, a thorough survey of techniques for conducting the associated tests and comparing distributions is provided.

**Strengths:**

The manuscript is well written, clearly describes related work, and offers a wealth of information regarding how to test for performance drift using hypothesis tests.

**Weaknesses:**

I found Figure 1 difficult to understand. In particular, I didn’t understand how the colors corresponded to covariate drift/concept shift, and how model recalibration fixed the problem.

In addition, while it is helpful to have such a detailed summary of tests, are some techniques more commonly used than others? It would be helpful to have more guidance regarding which methods are standard for ease of comparison.

**Questions:**

Does the variability in patients/clinical readers impact degradation performance testing? In particular, how challenging is it to modify any of the offered tests to success criteria that depends on 95% confidence intervals as opposed to point estimates?

In model performance monitoring (and perhaps elsewhere in the manuscript), the hypothesis test is defined with respect to a positive difference (tau). However, this may not be needed/feasible. In particular, it would be valuable to distinguish super-superiority, superiority, and non-inferiority as potential outcomes of the hypothesis tests.

**Alternative Position:**

Yes, and alternative positions are well-considered and addressed by the argument

**Author Identification:**

No.

**Context:**

4

**Discussion:**

4

**Ethics:**

["NO or VERY MINOR ethics concerns only"]

**Position:**

Yes, the paper argues for or against a position related to machine learning.

**Support:**

4

**Thoroughness:**

3

---

### Official Review · Reviewer_YiLC · 2025-08-08

**Significance:** 3
**Presentation:** 3
**Rating:** 6
**Confidence:** 4

**Summary:**

The authors argue that statistically valid, label-efficient post-deployment monitoring should be standard practice for clinical AI systems. The paper outlines why monitoring is essential for safety and describes common targets such as covariate shift, concept drift and performance degradation. The authors highlight regulatory expectations and identify key challenges, including costly labelling and operational constraints in healthcare settings.

**Strengths:**

1. The authors have a clear position “statistically valid, label-efficient post-deployment monitoring should be standard in clinical AI”), which is well-defined, tied to FDA/NIST guidance and immediately relevant to the healthcare ML community
2. Makes use of formal statistical testing frameworks (e.g., two-sample tests, sequential analysis, bootstrap) and applies them to operational ML in a clinical context which is something ML research and deployment (surprisingly) often lacks
3. I like how it balances theory with realities such as label cost, clinician alert fatigue and healthcare workflow limitations, which makes the recommendations credible

**Weaknesses:**

1. While individual concepts are well-described the paper often lists methods and open questions without deeply integrating them into a coherent positional narrative. This makes the argument less direct and those parts read more like a survey/review paper
2. The argument would be stronger with an end-to-end clinical example showing how the proposed principles translate into thresholds, alerting protocols and decision-making under uncertainty, i.e. case studies
3. The regulatory and statistical references are strong but I feel the paper under-cites recent advances in ML-specific drift detection (e.g., distribution-free detectors, adaptive control of false alarms) that could strengthen or challenge its stance. However this isn't a critical issue
4. The emphasis on i.i.d.-style statistical tests and straightforward two-sample testing glosses over the complexity of temporally correlated irregular and nonstationary clinical data
5. The framing risks implying that a method is either “valid” or “invalid” without acknowledging spectrum-of-validity trade-offs. I wonder if there is an angle here to relate this to the question of fairness vs accuracy, e.g. Kleinberg et al. Inherent trade-offs in the fair determination of risk scores (2017)

**Questions:**

1. If statistical validity, label efficiency, and subgroup monitoring cannot all be fully satisfied due to operational constraints, how would the authors prioritise among these goals, and what criteria would they use to decide?
2. Once a statistically valid alert is triggered, what is the authors’ view on how that should feed back into model improvement or retraining workflows without compromising future validity checks? Should the model be pulled? Or does the backup then become part of the equation?
3. Could the authors comment on whether monitoring systems should also track downstream clinical impact metrics (e.g., changes in treatment patterns, patient outcomes) alongside statistical drift metrics, and how that would fit in their framework?

**Alternative Position:**

Yes, and alternative positions are well-considered and addressed by the argument

**Author Identification:**

No.

**Context:**

3

**Discussion:**

3

**Ethics:**

["NO or VERY MINOR ethics concerns only"]

**Position:**

Yes, the paper argues for or against a position related to machine learning.

**Support:**

3

**Thoroughness:**

4

---

### Note · Authors · 2025-09-04

**1-11 Submit Again:**

Definitely yes

**1-1 Submission Process:**

4

**1-2 Next Year:**

A more specific track rubric with a couple of exemplars, and a more structured author–reviewer dialogue. Clearer logistics—earlier milestone dates and standardized forms—would also be really helpful, and a brief post-cycle summary of anonymized stats and common pitfalls could guide authors next year.

**1-3 Future Development:**

To make the track feel like an ongoing dialogue.

**1-4 Interest:**

["Panel discussions with other position paper authors", "Structured debates on controversial topics", "Mentorship programs for early-career researchers"]

**1-5 Thoughtful:**

9

**1-6 Supportive:**

8

**1-7 Technical Aspects Versus Position:**

2

**1-8 Gate Keeping:**

10

**1-9 Camera Ready Changes:**

1) Add a running example throughout the paper to strengthen the narrative.
2) Expand more on guidance regarding tests/methods, to clarify which techniques are more commonly used than others and in what scenarios.
3) Expand more on related work coverage.
3.1) Incorporate more ML-specific drift detection (e.g., distribution-free detectors, adaptive false alarm control) and clarify how our framework complements these methods.
3.2) Add a short paragraph in the Introduction and Limitations clarifying the Pareto trade-offs (false alarms, detection delay, label budget, subgroup considerations), noting the connection to fairness–accuracy trade-offs (Kleinberg et al., 2017)
4) Clarify Figure 1 (colors and figure description)
5) Clarify that data is i.i.d. only “locally” (during pre-deployment or post-deployment)
6) Add information on high-dimensional and multimodal settings
7) Add information on confounding variables or unobserved shifts in label semantics

**3-1 Review Response1:**

YiLC

**3-2 Reaction To Review1:**

We thank the reviewer for their thoughtful and constructive feedback. We appreciate the recognition of our central position and the balance of statistical rigor with clinical realities.

On narrative coherence: Thank you for the note. In the camera-ready, we will add a running clinical example (introduced in the Introduction and carried through Sections 3–5).

On related work. We will cite recent ML-specific drift detection advances (e.g., distribution-free and adaptive detectors) and clarify how our work complements rather than replaces these methods.

On data assumptions: Our framework compares two periods—pre (t_0) and post (t_1)—without assuming i.i.d. across time; the only i.i.d. assumption is local within each period (samples near t_0 or t_1). We acknowledge this is an approximation that omits richer temporal dependence; extending to dependence-aware/time-series change detection is a natural direction for future work, which we will note explicitly.

Our prioritization (1) validity, then (2) label efficiency, then (3) risk-based subgroup monitoring is our recommended operational policy and is consistent with the risk-based emphasis in NIST/FDA guidance.

Post-alert integration and outcomes: We prefer structured escalation rather than automatic retirement. When an alert is validated, we (1) apply immediate safeguards or a temporary switch to a validated backup; (2) run evaluations of potential updates while the primary remains constrained; and (3) pursue targeted remediation via active labeling of high-value cases and focused retraining. To preserve future validity, we pre-register triggers, keep monitoring/alert labels quarantined from model selection, and use out-of-time/nested holdouts. While this paper scopes to data/model monitoring, mature deployments should also track downstream clinical outcomes; we flag that as follow-up work.

**3-3 Review Response2:**

ZCZH

**3-4 Reaction To Review2:**

We thank the reviewer for their constructive feedback and for highlighting the clarity of the writing and comprehensiveness of the statistical testing overview.

On Figure 1. We appreciate the feedback and will revise the caption and labeling to clarify how colors denote covariate drift vs. concept shift, and how recalibration addresses miscalibration rather than full concept drift.

On guidance regarding methods. We agree it would help to indicate which tests are most widely used in practice. We will add a short synthesis table highlighting which techniques are standard, which are emerging, and how they compare in terms of ease of deployment.

On the impact of patient/reader variability: Yes, variability can inflate apparent drift. Tests based on confidence intervals (rather than point estimates) are straightforward extensions and provide a more robust basis for degradation criteria; we will clarify this.

On outcomes of tests: Thank you for the feedback. We can extend the framework to report three “positive” outcomes plus a fail state. We can predefine two clinical margins—a minimum meaningful improvement (for super-superiority) and a largest acceptable loss (for non-inferiority)—and, at each monitoring cycle, compute a one-sided lower confidence bound for the pre- vs post-difference using the appropriate method for the metric. Decisions can then follow a single ladder: if the lower bound exceeds the improvement margin, conclude super-superiority; if it is above zero, conclude superiority; if it remains above the acceptable-loss margin, conclude non-inferiority; otherwise, classify as degradation. Specification floors (and, if used, equivalence tests) remain hard guardrails—failing them is non-acceptable regardless.

We thank the reviewer again for their helpful suggestions, which will improve both the clarity and practical guidance of the paper.

**3-5 Review Response3:**

8KGL

**3-6 Reaction To Review3:**

We thank the reviewer for their thoughtful and encouraging assessment. We appreciate the recognition of our framing of monitoring as hypothesis testing, and the positive feedback on our treatment of label scarcity and subgroup degradation.

On data assumptions: Our framework compares two periods—pre (t_0) and post (t_1)—without assuming i.i.d. across time; the only i.i.d. assumption is local within each period (samples near t_0 or t_1). We acknowledge this is an approximation that omits richer temporal dependence; extending to dependence-aware/time-series change detection is a natural direction for future work, which we will note explicitly.

On high-dimensional and multimodal data: Our framework is modality-agnostic—it can operate on learned embeddings and apply the same drift, performance, and subgroup tests. We also note the trade-off that higher dimensionality raises sample requirements and can reduce power, so label-efficient tactics (such as sequential looks and active labeling) and clear reporting of margins/thresholds are key. We will add this information in the camera-ready version.

On confounding and unobserved label shifts: We treat both as forms of concept drift and test for them by comparing the joint pre- vs post-deployment distribution of features, clinical/demographic variables, and labels. When drift or degradation is detected, we follow with subgroup identification, root-cause analysis, and mitigation (e.g., re-weighting/debiasing, targeted retraining). For label semantics, we rely on surrogate labels (e.g., ICD codes, labs) when ground truth is scarce, while noting these proxies can be noisy and themselves drift—so they are monitored within the framework. A fully causal treatment is beyond scope; in the camera-ready, we will state these limits explicitly and add guidance on monitoring surrogate-label drift.

---

### Meta-Review · Area_Chair_xdJH · 2025-08-29

**Rating:** 7
**Confidence:** 5

**Strengths:**

I quickly read the paper and agree with reviewers comments

- well written, clearly describes related work, and offers a wealth of information regarding how to test for performance drift using hypothesis tests.

- balances theory with realities such as label cost, clinician alert fatigue and healthcare workflow limitations, which makes the recommendations credible

**Weaknesses:**

Some reviewers proposed meaningful comments and I think most do not constitute significant concerns regarding the acceptance. But the followings are worthy to revise

- the authors acknowledge the high dimensionality of clinical data, practical strategies for selecting and validating appropriate test statistics in such settings are underexplored. The reliance on statistical tests may also struggle with the complexity and multimodality of clinical feature spaces, especially when concept drift and covariate shift co-occur.

- In model performance monitoring (and perhaps elsewhere in the manuscript), the hypothesis test is defined with respect to a positive difference (tau). However, this may not be needed/feasible. In particular, it would be valuable to distinguish super-superiority, superiority, and non-inferiority as potential outcomes of the hypothesis tests.

- Could the authors comment on whether monitoring systems should also track downstream clinical impact metrics (e.g., changes in treatment patterns, patient outcomes) alongside statistical drift metrics, and how that would fit in their framework?

**Questions:**

In general, I have no major concerns to improve the paper, I would suggest a clarification in the variable ''C''

**Thoroughness:**

2

---

### Decision · Program_Chairs · 2025-09-26

Accept